# Cross-sectional study of prevalence, causes and trends in visual impairment in Nirmal District, Telangana, India: Nirmal Eye Evaluation for Trends study

Srinivas Marmamula [1,2,3,4] Aritra Chinya,[1] Vijay Kumar Yelagondula [3] Rajashekar Varada,[1] Rohit C Khanna [1,2,5] Raja Narayanan[4,6]

**Correspondence to**
Dr Srinivas Marmamula;
sri.marmamula@lvpei.org

## ABSTRACT

**Objective** To determine the prevalence, causes and risk factors associated with visual impairment (VI) in the Nirmal district of Telangana, India, using extended Rapid Assessment of Visual Impairment (RAVI) methodology.

**Design** Cross-sectional study.

**Setting** Community setting.

**Participants** Participants aged ≥16 years were enumerated from 90 randomly selected clusters and 4629/5400 (85.7%) participants were examined. Presenting visual acuity (VA) was assessed using a Snellen chart with E optotypes at a 6 m distance. Near vision was assessed binocularly using an N notation chart with tumbling E optotypes at a 40 cm distance. An anterior segment examination done followed by distance direct ophthalmoscopy at 50 cm. Non-mydriatic fundus images were obtained. VI was defined as presenting VA worse than 6/12 in the better eye. The prevalence of VI in the current study was compared with a RAVI study conducted in 2014 to assess the trends in VI among those aged ≥40 years.

**Primary outcome** Prevalence, causes and risk factors for VI.

**Results** Among those examined, 55% were women, 53% had at least school-level education, 2.3% self-reported diabetes and 8.7% self-reported hypertension. The prevalence of VI was 8.81% (95% CI 8.01% to 9.67%). Overall, uncorrected refractive errors (49.5%) were the leading cause of VI, followed by cataracts (40.2%) and posterior segment diseases (4.9%). Among those aged ≥40 years, the prevalence of VI declined by 19.3% compared with the 2014 baseline study (from 20.2% to 16.3%; p<0.01).

**Conclusion** The extended RAVI study conducted in the Nirmal district showed a considerable decline in the prevalence of VI. Targeted interventions are needed to provide adequate eye care for the high-risk groups in this district.

## INTRODUCTION

Over a billion people suffer due vision impairment (VI) globally, with cataracts and uncorrected refractive errors (URE) being the leading causes.[1 2] Both these conditions can be addressed using cost-effective interventions, such as spectacles and cataract surgery.

## STRENGTHS AND LIMITATIONS OF THIS STUDY

⇒ Rapid assessments typically focus on participants aged ≥40 years. This study extends the rapid assessment methodology to include younger age groups (≥16 to 39 years) and provides estimates on the prevalence and causes of visual impairment (VI).

⇒ In addition to prevalence estimates, temporal trends in the prevalence of VI are presented.

⇒ As a randomly selected population-based sample was used, the results from the study can be extrapolated to the population in the region.

⇒ The over-representation of women could have overestimated the prevalence of VI in our study.

Population-based data on the prevalence and causes of VI are essential to plan eye care service delivery models to address this global problem. Though conventional epidemiological studies provide the data, they are often resource-intensive and need expertise to implement them. The rapid assessment methods are low-cost epidemiological tools that provide data on the prevalence and causes of VI using limited resources while being relatively easy to implement. In addition, these rapid assessments can be repeated at stipulated intervals to study the temporal trends in a given region.[3] Rapid assessment studies are even more important now, with WHO setting global targets for effective cataract surgical coverage and effective refractive error coverage as indicators to measure the progress towards universal eye health.[4]

Rapid assessment studies initially focused on cataract alone; however, they were modified and evolved to cover other causes of VI, with an increasing focus on emerging eye conditions, such as diabetic retinopathy and refractive errors.[3 5] Rapid Assessment of Visual Impairment (RAVI) is the offshoot of multiple rapid assessment methods developed and has been used extensively in India

and other countries.[6–11] Studies using the RAVI methodology focus on individuals aged 40 years and older. Recently, it has been modified to include younger individuals (≥16 years) and has been renamed as the extended RAVI methodology.[12] In addition, new tools have been added to collect data on systemic conditions and disabilities, helping to more holistic planning of eye health programmes.[5 13] The Nirmal Eye Evaluation for Trends is the first study to use extended RAVI. In this study, we report the prevalence, causes and risk factors of VI in the Nirmal district and adjoining areas of Telangana, India. In addition, this paper also compares the temporal trends in the prevalence of VI in this region using data from a previous study conducted in 2014.[14]

## MATERIALS AND METHODS

### Patient and public involvement
Patients and other members of the public were not involved in the design of the study.

### Sampling strategy
Assuming a VI (presenting visual acuity (VA) worse than 6/12) prevalence of 3.5%, allowing for a 95% CI, a precision of 20%, a design effect of 1.6 for a predetermined cluster size of 60 participants and a 20% non-response rate, the minimum sample size required was 5270, which was rounded up to 5400 participants (90 clusters). A multistage cluster random sampling procedure with a compact segment sampling method was used, which has been described in previous reports.[14] The study area had a population of 0.5 million people and comprised 10 subdistricts (mandal) from the Nirmal (8) and Nizamabad (2) districts. The eye care needs of the study area were serviced by a secondary centre of L V Prasad Eye Institute. Data were collected between November 2021 and March 2022.

### Data collection
Three teams, each comprising a vision technician and two community eye health workers, collected the data. They were supervised by a study coordinator (optometrist), who was also responsible for travel logistics and quality control. The examiners were trained to conduct the study procedures and document the findings. A reliability assessment was conducted before the study to assess the interobserver agreement on VA with a gold-standard senior optometrist. All examiners had a good agreement with the gold-standard optometrist (kappa 0.8 or more).

One of the three study teams visited participants at their homes and conducted eye examinations. The time of the visits was planned to maximise the availability of the participants at their households for examination. In each selected household, all the individuals who fulfilled the age criteria were documented, and all those who were available during the visit were examined. At least two attempts were made to examine those who were

unavailable during the first visit, after which they were marked as unavailable.

### Eye examination protocol
A brief interview was conducted to collect personal and sociodemographic information, such as age, education level and systemic health conditions (online supplemental file 1). Data related to the ocular history, including current and previous use of spectacles, use of eye-drops and details of any previous surgery, were also collected. Information regarding the barriers to the uptake of eye care services was also collected using a structured questionnaire.

The standard RAVI clinical examination was conducted after the interview, as described in previous studies.[11 15–17] In brief, the distance visual acuity (VA) was assessed using a standard Snellen chart with tumbling E optotypes at a distance of 6 m. If a participant was unable to identify the letters in the first line of the chart, the distance between the participant and the chart was progressively reduced to 3 m and then 1 m till VA could be recorded. Unaided VA was recorded for all participants. Aided VA was recorded for participants using spectacles for correction. Aided VA was considered as the presenting VA (PVA) for those with spectacles, and an unaided VA was considered as the PVA for those without spectacles. If the PVA was worse than 6/12, the VA was recorded using a multiple pinhole occluder. Near vision was assessed binocularly using the N notation near vision chart at a fixed distance of 40 cm in ambient lighting conditions. The fixed distance was maintained using a string attached to the near vision chart. Both unaided and aided near vision were assessed if the participant reported spectacles use for near. Near vision was reassessed using near addition lenses in a trial frame appropriate for that age among participants with near vision worse than N6.

An eye examination was performed using a torchlight/portable handheld slit lamp (Keeler PSL Classic, USA). The lens was assessed using distant direct ophthalmoscopy at approximately 50 cm distance in a shaded area (indoors), which was graded as normal, obvious lens opacity, aphakia, pseudophakia without posterior capsular opacification (PCO) or pseudophakia with PCO. If the lens could not be examined because of corneal opacities, phthisis bulbi or absent globe, then it was documented in the data form. A non-mydriatic portable fundus camera (Visuscout 100 Handheld Fundus Camera, Carl Zeiss Meditec, USA) was used to capture retinal images. Two images, one optic disc-centred and another macula-centred, were captured for each eye. All the images were evaluated by experienced graders at L V Prasad Eye Institute. The participants with VI and those requiring other eye care services were referred to the nearest eye care facility for management.

The WHO categorises visual impairment (VI) into four categories based on the presenting VA in the better eye.[18] The four categories are as follows: mild VI (MiVI—VA worse than 6/12–6/18), moderate VI (MVI—VA worse

than 6/18–6/60), severe VI (SVI—worse than 6/60–3/60) and blindness (VA worse than 3/60 to no perception of light). The case definitions for the causes of VI used in this study have been described in our previous publications.[11] In brief, URE was defined as presenting VA<6/12, improving to 6/12 or better with pinhole. Cataract was defined as an opacity of the crystalline lens as seen with torchlight and obscuring the red reflex, partially or completely, on the distance direct ophthalmoscopy, resulting in a VA<6/12 that does not improve with pinhole. Posterior segment disease was considered as the cause of VI in cases where there was no media opacity and VA did not improve with a pinhole. Posterior Capsular Opacification (PCO), phthisis bulbi/absent globle and corneal opacities/oedema after cataract surgery were marked as surgical complications. After the eye examination, the principal cause of VI was recorded for each eye separately, and then for the person. If there was more than one cause, the cause that was more easily treatable or correctable was marked as the main cause of VI.

As this study was conducted during the pandemic, all COVID-19-related protocols were followed, including the use of masks (N-95) and visors at all times, frequent hand sanitisation and social distancing. All the team members were vaccinated before the start of the study. The equipment used, such as trial frames and multiple pinhole occluders, were disinfected with alcohol wipes/swabs after each use. The participants were offered hand sanitiser to clean their hands before starting the study procedures. The current health status of all the participants was enquired before the eye examinations.

### Data management
In the field, data were collected using paper forms. The forms were then transported to the data centre for entry into a Microsoft Access database. Data analyses were performed using the Stata Statistical Software for Windows, V.14 (StataCorp). The prevalence estimates were adjusted to the age and gender distribution of the population for the year 2011 and presented with the 95% CIs. The demographic associations of VI with age, gender, education and systemic conditions were assessed using multiple logistic regression models and adjusted ORs with 95% CI are reported. A study using the conventional RAVI methodology (included individuals aged 40 years and older) was conducted in the same region in 2014.[14] The prevalence estimates from the current study were compared with the 2014 study to assess the trends in VI over time in this region.

### RESULTS
### Characteristics of the participants
Of the 5400 participants included in this study from the 90 clusters, 4629 (85.7%) were examined. The mean age (±SD) of the examined participants was similar to those not examined (42.5 (±16.6) years vs 42.0 (±16.5) years; p=0.38). A higher proportion of women were examined

**Table 1** Visual impairment (VI) and demographic characteristics of the participants

|  | Participants examined (n) | Participants with VI n (%) | P value |
|---|---|---|---|
| Age group (years) |  |  | <0.01 |
| 16–29 | 1244 | 12 (1.0) |  |
| 30–39 | 1010 | 12 (1.2) |  |
| 40–49 | 899 | 30 (3.3) |  |
| 50–59 | 684 | 81 (11.8) |  |
| 60–69 | 440 | 127 (28.9) |  |
| 70 and above | 352 | 146 (41.5) |  |
| Gender |  |  | 0.11 |
| Male | 2084 | 199 (9.5) |  |
| Female | 2545 | 209 (8.2) |  |
| Education level |  |  | <0.01 |
| No education | 2173 | 356 (16.4) |  |
| Any education | 2456 | 52 (2.1) |  |
| Diabetes |  |  | <0.01 |
| Yes | 129 | 35 (27.1) |  |
| No | 4500 | 373 (8.3) |  |
| Hypertension |  |  | <0.01 |
| Yes | 402 | 73 (18.2) |  |
| No | 4227 | 335 (7.9) |  |
| Total | 4629 | 408 (8.8) |  |

(55% vs 44%; p<0.01). Among those examined, 55% (n=2545) were women, 53% (2456) had at least school education, 2.3% (n=129) self-reported diabetes and 8.7% (n=402) self-reported hypertension (table 1).

The overall crude prevalence of VI was 8.81% (95% CI 8.01% to 9.67%). This included MiVI (2.87%; 95% CI 2.41% to 3.40%), MVI (4.6%; 95% CI 4.06% to 5.29%), SVI (0.60%; 95% CI 0.40% to 0.87%) and blindness (0.69%; 95% CI 0.47% to 0.97%). Age-adjusted and gender-adjusted prevalence was 7.15% (95% CI 6.80% to 8.32%) (table 2).

**Table 2** Crude prevalence and age-adjusted and gender-adjusted prevalence of visual impairment (VI)

|  | Crude prevalence of VI (95% CIs) | Age and gender-adjusted prevalence of VI (95% CIs) |
|---|---|---|
| Mild VI | 2.87 (2.41 to 3.40) | 2.37 (2.00 to 2.90) |
| Moderate VI | 4.64 (4.06 to 5.29) | 4.03 (3.50 to 4.64) |
| Severe VI | 0.60 (0.40 to 0.87) | 0.55 (0.35 to 0.80) |
| Blind | 0.69 (0.47 to 0.97) | 0.57 (0.37 to 0.82) |
| All VI | 8.81 (8.01 to 9.67) | 7.51 (6.80 to 8.32) |

### Risk factors for VI

On univariate analysis, the prevalence of VI was highest (41.2%) in the oldest age group. Though the prevalence of VI did not vary with gender (p=0.11), it was significantly higher among those with no formal education (16.4% vs 2.1%; p<0.01). The prevalence of VI was also higher among those who self-reported hypertension (18.2% vs 7.9%; p<0.01) and diabetes (27.1% vs 8.3%; p<0.01).

The multiple regression analysis showed that the odds for VI increased with increasing age. Compared with the participants aged 16–29 years, the odds for VI were 6.78 (95% CI 3.46 to 13.29) for the 50–59 age group, 6.7 (95% CI 3.4 to 13.2) for the 60–69 age group and 33.7 (95% CI 17.19 to 66.37) in the above 70 and older age group. The participants with no formal education had higher odds for VI compared with those with formal education (OR 2.75; 95% CI 1.90 to 3.97). Similarly, the participants with a history of diabetes had higher odds for VI (OR 1.83; 95% CI 1.16 to 2.89). Women had lower odds for VI compared with men (OR 0.64; 95% CI 0.51 to 0.82). Hypertension was not associated with VI (p=0.321) (table 3).

### Causes of VI

Overall, UREs (49.5%) were the leading cause of VI, followed by cataract (40.2%) and posterior segment diseases (4.9%). UREs were the leading cause of moderate and severe VI, and cataract was the leading cause of blindness. UREs were the leading cause of VI in the younger age group (16–59 years), and cataract was the leading cause of VI in the older age group (60 years and older) (figure 1).

**Table 3** Effects of sociodemographic variables on visual impairment (multiple logistic regression analysis)

|  | Adjusted OR (95% CI) | P value |
|---|---|---|
| **Age group (years)** | | |
| 29 | Reference | |
| 30–39 | 0.93 (0.4 to 2.11) | 0.865 |
| 40–49 | 2.02 (0.99 to 4.13) | 0.053 |
| 50–59 | 6.78 (3.46 to 13.29) | <0.01 |
| 60–69 | 18.83 (9.6 to 36.87) | <0.01 |
| ≥70 | 33.77 (17.19 to 66.37) | <0.01 |
| **Gender** | | |
| Male | Reference | |
| Female | 0.64 (0.51 to 0.82) | <0.01 |
| **Education** | | |
| Any education | Reference | |
| No education | 2.75 (1.90 to 3.97) | <0.01 |
| **Hypertension** | | |
| No | Reference | |
| Yes | 0.85 (0.62 to 1.17) | 0.321 |
| **Diabetes** | | |
| No | Reference | |
| Yes | 1.83 (1.16 to 2.89) | 0.01 |

### Temporal trends in VI

The data of individuals aged ≥40 years examined in the 2014 and 2021–2022 studies were analysed to capture the

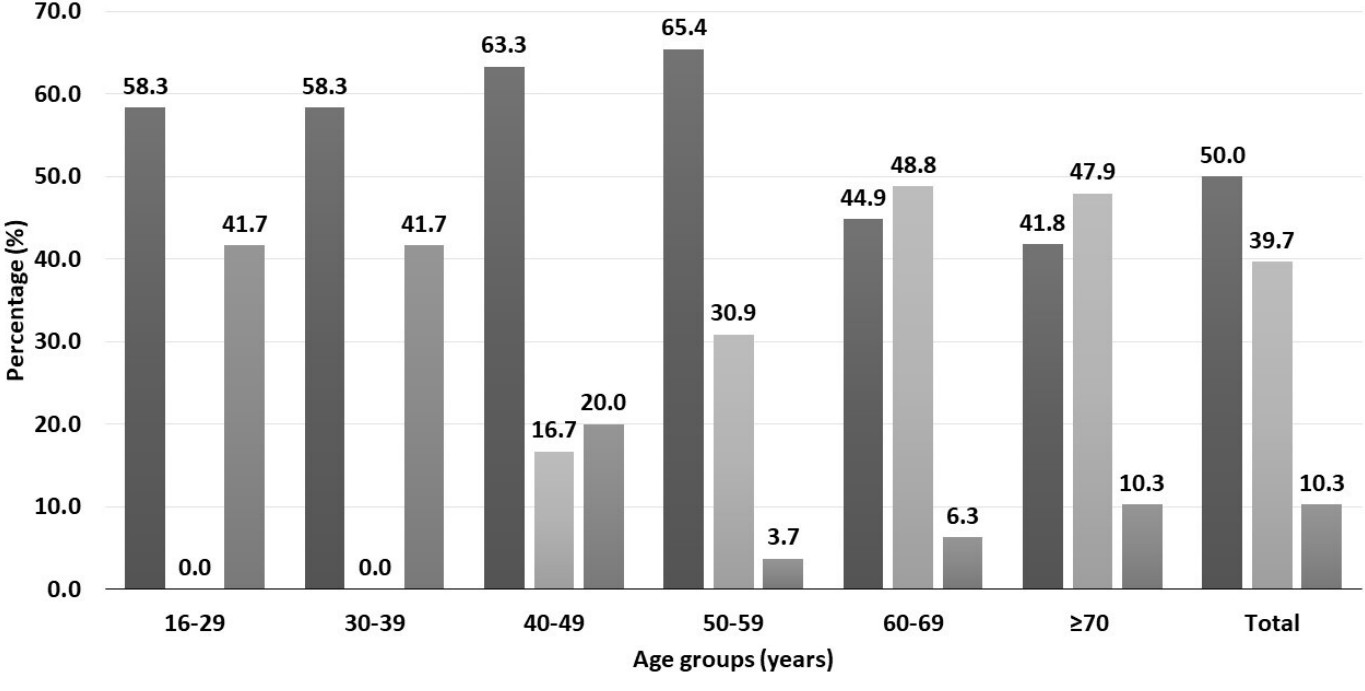

**Figure 1** Causes of visual impairment across the age groups.

trends in the prevalence of VI over time.[14] In total, 2974 and 2375 participants aged ≥40 years were examined in the 2014 and 2021–2022 studies, respectively. The mean age (±SD) of the participants was higher in the 2022 study (55.0±11.0 years vs 51.7±9.9 years; p<0.05). Similarly, a lesser proportion of men were examined in the 2022 study (42.5% vs 45.6%; p=0.03). Overall, the prevalence of VI declined by 19.3% compared with the 2014 baseline study (from 20.2% to 16.3%; p<0.01). In terms of categories of VI, MiVI declined by 21.9% (from 6.4% to 5.0%; p=0.03), MSVI declined by 15.1% (from 11.9% to 10.1%; p=0.04) and blindness declined by 36.8% (from 1.9% to 1.6%).

## DISCUSSION

We have reported on the prevalence and causes of VI among the adult population in the Nirmal district of Telangana using the extended RAVI methodology. The conventional RAAB and RAVI methods include individuals aged ≥50 years and ≥40 years, respectively. In contrast, the extended RAVI methodology used in this study included anyone aged ≥16 years. While it is advantageous to only include the older population to minimise the sample size and use of resources, the data on VI are not readily available in the younger age groups. Data on all ages are essential to plan universal eye health initiatives in the region. The extended RAVI is an attempt to provide comprehensive information on the prevalence of VI in the complete adult population in this region. The data from this study can supplement the data from school eye health programmes, providing a complete picture of VI in the entire population, other than children under 5 years. In addition, we used the revised WHO definitions in this study for cross-comparison with other studies done in India and other regions of the world.

The Andhra Pradesh Eye Disease Study (APEDS) conducted between 1996 and 2000 was the only population-based cross-sectional study that included the population of all ages. The prevalence of moderate VI, severe VI and blindness was 10.1%, 2.3% and 2.3%, respectively.[19 20] Using similar definitions, the prevalence of moderate VI, severe VI and blindness in this study was 4.6%, 0.60% and 0.69%, respectively.[19 21] The prevalence of Mild VI is not reported in APEDS. Despite a difference in the age groups between the studies, a lower prevalence in this study indicates a decline in the prevalence of VI in this region over the last three decades. Such a secular trend of decline in VI has been reported from various locations, suggesting an improvement in the availability and uptake of eye care services in this region.

Both APEDS and the current study had a higher prevalence of VI among the older participants, which is consistent across all the studies conducted in this region.[19 21] In this study, though the prevalence did not vary with gender, women had lower odds for VI, which is contrary to the APEDS study, where women had a higher prevalence of VI. This difference could be attributed to the availability, acceptability and a higher uptake of eye care services among women. A higher prevalence of VI was also noted among those with lower levels of education, which is similar to other studies in this region.[11 14 19 21] The higher visual needs and availability of resources for eye examinations and treatment might have contributed to a lower prevalence of VI among those with higher levels of education. The participants who self-reported diabetes had a higher prevalence of VI in this study, which might be caused due to an earlier onset of cataract secondary to diabetes and other refractive changes in the eye.

Several studies have reported the prevalence of VI using the RAVI methodology among participants aged

**Table 4** Prevalence of visual impairment in Nirmal district in Telanagana, India as reported in various studies

| | Year | Sample size | Moderate visual impairment (presenting visual acuity worse than 6/18–6/60) | Severe visual impairment (presenting visual acuity worse than 6/60–3/60) | Blindness (presenting visual acuity worse than 3/60) | Total visual impairment (presenting visual acuity worse than 6/18) |
|---|---|---|---|---|---|---|
| | | n | % | % | % | % |
| ≥40 years | | | | | | |
| APEDS[19 20] | 1997–1998 | 840 | 28.5 | 1.8 | 5.0 | 35.2 |
| RAVI[14] | 2014 | 2974 | 10.4 | 1.5 | 1.9 | 13.7 |
| RAVI | 2022 | 2392 | 8.8 | 0.4 | 0.6 | 9.7 |
| ≥50 years | | | | | | |
| APEDS[19 20] | 1997–1998 | 521 | 40.3 | 2.9 | 7.5 | 50.7 |
| RACSS[27] | 2006–2007 | 2160 | 13.6 | 4.8 | 3.2 | 21.6 |
| RAVI[14] | 2014 | 1550 | 16.6 | 2.3 | 3.0 | 21.9 |
| RAVI | 2022 | 1491 | 13.4 | 1.1 | 1.7 | 16.2 |

APEDS, Andhra Pradesh Eye Disease Study; RACSS, Rapid Assessment of Cataract Surgical Services; RAVI, Rapid Assessment of Visual Impairment.

≥40 years in Andhra Pradesh, Telangana and other parts of India.[14 15 17–22 22 23 23 24] The prevalence of VI ranges from 8.7% in Tripura,[24] 10% in Krishna district in Andhra Pradesh,[15] 11.4% in Delhi,[22] 12.8% in Ganjam and Khordha districts in Odisha,[23] 12.8% in Akividu region West Godavari and Krishna districts[17] and 13.7% in Mahbubnagar and Adilabad districts in Telangana.[14] These are comparable to the 11.3% found in the current study.

UREs and cataract remain the leading causes of VI.[11 14 17 24–26] Similar to APEDS, UREs are the leading causes of VI in the older age groups compared with the younger age groups.[19 21] Moreover, similar to other studies, cataract was more common among those with severe grades of VI.[11 14 17 25 26] As both these conditions are manageable with cost-effective interventions, strategies are needed to reach these communities and provide eye care.

The temporal trends in the prevalence of VI have been reported from Telangana.[26–28] In an earlier paper, we compared two studies conducted using the same methodology and geographical locations, Khammam and Warangal districts, that included identical age groups.[26] There was a 2.5% decline in VI in Khammam district, but it remained stable in Warangal district over 5 years.[26] In this study, we observed a 19% decline in VI compared with the study conducted in 2014, which is an annual decline of 2%. This decline could be attributed to increased availability and uptake of eye care services in this region. However, due to the absence of a control arm, the role of secular trends resulting in the decline of VI cannot be ruled out.

The Nirmal district (erstwhile Adilabad district) has witnessed a few epidemiological studies over the years. APEDS was conducted in 1997–1998, followed by Rapid Assessment of Cataract Surgical Services (RACSS), a couple of RAVI studies in 2014, and the current study in 2022 (table 4). Among those aged ≥40 years, the prevalence of VI (<6/18 definition) was 35.2% in APEDS, which dropped to 13.7% in the 2014 RAVI study. The prevalence has declined further to 9.7% in the current study. Among those aged 50 years and older, the prevalence of VI (<6/18 definition) was 50.7% in APEDS, which dropped to 21.6% in RACSS conducted in 2006–2007 and remained stable in the 2014 RAVI study, declining to 16.2% in the current study. As different protocols have been used over the years, direct comparisons are limited by the different measurement methods. Nevertheless, VI is declining in this region, as indicated by the two recent RAVI studies using identical protocols.

A good response rate, a randomly selected population-based sample, the use of an updated WHO definition of VI, and the inclusion of participants of wider age groups are the strengths of this study. After APEDS, this is the major study to report VI in the adult population in this region. However, a few studies have reported the VI in older age groups during this gap of two decades. A higher proportion of women were examined in this study, which could be due to the migration of men to urban areas in search of work, a common occurrence in Telangana. The previous studies in this region also show a female preponderance.[11 14] Therefore, the over-representation of women could have overestimated the prevalence of VI in our study. Another inherent limitation of rapid assessment methods is the ascertainment of causes of VI. The major cause is considered based on the ease of remedy to address the VI. Often, the prevalence of cataract and refractive errors is overestimated as they are easy to treat and correct, respectively, compared with posterior segment conditions. This limitation applies to the current study as well as the rapid assessment methodology was used.

In conclusion, a significant burden of VI is observed in the region. However, the declining trend in the prevalence of VI suggests that the eye care services in the region are improving. This study can be a guide for more focused efforts to address vision loss and achieve universal eye health in this region. Also, there is a need for integration between eye care services and primary healthcare as people with diabetes had a higher prevalence of VI. VI impedes the attainment of sustainable development goals and the overall quality of life. Efforts to address vision loss might have a ripple effect on the overall health and well-being of individuals, families and communities, contributing towards sustainable development goals.

**Author affiliations**
[1]Allen Foster Community Eye Health Research Centre, Gullapalli Pratibha Rao International Centre for Advancement of Rural Eye care, L V Prasad Eye Institute, Hyderabad, India
[2]School of Optometry and Vision Science, University of New South Wales, Sydney, New South Wales, Australia
[3]Brien Holden Institute of Optometry and Vision Science, L V Prasad Eye Institute, Hyderabad, India
[4]Wellcome Trust / Department of Biotechnology, India Alliance, L V Prasad Eye Institute, Hyderabad, India
[5]School of Medicine and Dentistry, University of Rochester, New York City, New York, USA
[6]Anant Bajaj Retina Institute, L V Prasad Eye Institute, Hyderabad, India

**Acknowledgements** The authors thank the vision technicians (Sunil Anant Doke, Vaibhav Kumar Dave, Jalindar Kumar Doke, Degaon Ganesh, Gangadharolla Vamshi) and community eye health team (Yedla Sandeep, Akaram Chinnaiah, Laxman Sunkari, Gunde Rao Rahul, Ramnapoina Mahesh) for the assistance provided in data collection. The authors also thank Sandeep Dayal, Thirupathi Reddy Kumbham, Shekhar Konegari and Saptak Banerjee for monitoring the study. Abhinav Sekar is acknowledged for the language inputs on earlier versions of our manuscript. The authors thank the volunteers for their participation in the study.

**Contributors** SM conceived the idea, designed and conducted the study, analysed the data and wrote the manuscript. AC was involved in data collection and quality control. VKY, RV, RCK and RN reviewed the earlier version of the manuscripts and provided the intellectual inputs. SM is responsible for the overall content as a guarantor.

**Funding** This work was supported by the DBT Wellcome Trust India Alliance Clinical Research Centre Grant (grant number IA/CRC/19/1/610010) awarded to Dr Raja Narayanan and Hyderabad Eye Research Foundation, India.

**Competing interests** None declared.

**Patient and public involvement** Patients and/or the public were not involved in the design, or conduct, or reporting, or dissemination plans of this research.

**Patient consent for publication** Consent obtained directly from patient(s).

**Ethics approval** The study protocol was approved by the Institutional Review Board (IRB) of Hyderabad Eye Research Foundation, L V Prasad Eye Institute (LVPEI) (Reference ID: LEC-08173). This study was conducted in accordance with the tenets of the Declaration of Helsinki. Written informed consent was obtained from all the participants. For those aged less than 18 years of age, assent for the eye examination was obtained from the participant and written informed consent was obtained from the legal guardian.

**Provenance and peer review** Not commissioned; externally peer reviewed.

**Data availability statement** No data are available.

**ORCID iDs**
Srinivas Marmamula http://orcid.org/0000-0003-1716-9809
Vijay Kumar Yelagondula http://orcid.org/0009-0003-3983-8783
Rohit C Khanna http://orcid.org/0000-0002-8698-5562

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
