## [Reviewer comments · BMJ Open]

ARTICLE DETAILS

TITLE (PROVISIONAL)	A cross-sectional study of prevalence, causes, and trends in visual impairment in Nirmal District, Telangana, India - Nirmal Eye Evaluation for Trends study.
AUTHORS	Marmamula, Srinivas; Chinya, Aritra; Yelagondula, Vijay; Varada, Rajashekar; Khanna, Rohit; Narayanan, Raja

VERSION 1 – REVIEW

REVIEWER	H. Alrasheed, Saif Qassim University
REVIEW RETURNED	06-Jan-2024

GENERAL COMMENTS	Thank you for inviting me to review the article entitle “Prevalence, causes, and trends in Visual Impairment in Nirmal District, Telangana, India - Nirmal Eye Evaluation for Trends study.”. I have following comments and suggestions Abstract In method section, the authors mentioned “An anterior segment examination and distance direct ophthalmoscopy were also performed, and non-mydriatic fundus images were obtained”. At which distance, direct ophthalmoscopy was also performed. In method section, the authors mentioned “VI was defined as presenting VA worse than 6/12 in the better eye”. I think this definition is commonly used for children. For adults, presenting VI was defined as presenting VA worse than 6/18 in the better eye. Introduction Clear and informative Methods 1. The authors mentioned in page 5 lines 12-13 “The study participants were not involved in setting the research question or the outcome measures”. The above sentence means to me is unclear. 2. Page 5, line 23, why the authors use a 20% non-response rate? I think this higher non-response rate 3. Page 7, Lines 3-8 “The WHO categorizes visual impairment (VI) into four categories based on the presenting visual acuity in the better eye.[18] The four categories are as follows: Mild VI (MiVI - VA worse than 6/12 to 6/18), Moderate VI (MVI - VA worse than 6/18 to 6/60), Severe VI (SVI - worse than 6/60 to 3/60), and Blindness (VA worse than 3/60 to no perception of light)”. This definition of VI, it is better to move to the introduction section. Results Page 10-lines 50-60 section “Temporal trends in VI.”
---

	Please add the reference number of the study conducted in 2014 that was used for comparison. Discussion Clear and informative
REVIEWER	Lichter , Myrna University of Toronto I also conduct projects in Canada on social determinants of eye health, predominantly among homeless, refugee and Indigenous patients.
REVIEW RETURNED	11-Jan-2024
GENERAL COMMENTS	This was a very clear comprehensive study on a randomized selection of patients. There were some variables that may have been difficult to control, such as distant vision acuity testing in the subject's home. There may have been ambient light differences, or obstruction. I think this is a very minor point. The VI is higher that we have experienced in Canada (4%) but I think the sample size used here was larger. The results were not unexpected; leading cause of VI was uncorrected refractive error followed by cataract. I inferred from the paper that issues like spectacle correction were dealt with in an expedient manner, and referral for cataract surgery made. In all, an admirable reproducible study that should lead to better interventions and outcomes for patients. well done.

VERSION 1 – AUTHOR RESPONSE

Reviewer: 1 - Dr. Saif H. Alrasheed, Qassim University, Qassim University

Comments to the Author:

Thank you for inviting me to review the article entitle “Prevalence, causes, and trends in Visual Impairment in Nirmal District, Telangana, India - Nirmal Eye Evaluation for Trends study.”.

I have following comments and suggestions

Abstract

In method section, the authors mentioned “An anterior segment examination and distance direct ophthalmoscopy were also performed, and non-mydratic fundus images were obtained”.

At which distance, direct ophthalmoscopy was also performed.

Response: Distance direct ophthalmoscopy was done at approximately 50 cm in a semi-dark condition (indoors). We have included this in our revision.

In method section, the authors mentioned “VI was defined as presenting VA worse than 6/12 in the better eye”. I think this definition is commonly used for children. For adults, presenting VI was defined as presenting VA worse than 6/18 in the better eye.

Response: The revised definition of VI as per the ICD-11 classification is based on presenting visual acuity worse than 6/12 in the better eye. Kindly find the reference for this definition.

<https://icd.who.int/browse/2024-01/mms/en#1103667651>

WHO - ICD -11 Vision Impairment classification. Available <https://www.who.int/news-room/fact-sheets/detail/blindness-and-visual-impairment>; (Accessed 16 May 2023).

Introduction

Clear and informative

Response: Thank you for the positive comment.

Methods

1. The authors mentioned in page 5 lines 12-13 “The study participants were not involved in setting the research question or the outcome measures”.

The above sentence means to me is unclear.

Response: Thank you for pointing out this error. We have corrected this in our revision.

2. Page 5, line 23, why the authors use a 20% non-response rate? I think this higher non-response rate

Response: Thank you for this comment. We have been working in these populations for several years and based on our experience we have taken a non-response rate of 20%. We have had a similar response rate in our previous studies. As our study included a younger population and hence a higher level of non-response was anticipated. As results indicate we achieved, 86% response rate.

3. Page 7, Lines 3-8 “The WHO categorizes visual impairment (VI) into four categories based on the presenting visual acuity in the better eye.[18] The four categories are as follows: Mild VI (MiVI - VA worse than 6/12 to 6/18), Moderate VI (MVI - VA worse than 6/18 to 6/60), Severe VI (SVI - worse than 6/60 to 3/60), and Blindness (VA worse than 3/60 to no perception of light)”. This definition of VI, it is better to move to the introduction section.

Response: Thank you for the suggestion. As we wanted to define describing the study procedures for easy flow of information, we have included the definitions in the methods section. We prefer to retain the definition in the methods section.

Results

Page 10-lines 50-60 section “Temporal trends in VI.”

Please add the reference number of the study conducted in 2014 that was used for comparison.

Response: Thank you. We have added the reference to this 2014 study as suggested.

Discussion

Clear and informative.

Response: Thank you for the positive comments.

We are grateful for the valuable inputs to improve our manuscript.

Reviewer: 2

Dr. Myrna Lichter , University of Toronto

Comments to the Author:

This was a very clear comprehensive study on a randomized selection of patients. There were some variables that may have been difficult to control, such as distant vision acuity testing in the subject's

home. There may have been ambient light differences, or obstruction. I think this is a very minor point.

Response: Thank you for the positive comments and understanding of the study conditions.

The VI is higher than we have experienced in Canada (4%) but I think the sample size used here was larger.

Response: Thank you. Visual impairment depends on several factors including those related to availability and uptake of services. This could partly explain the difference in VI prevalence across the regions.

The results were not unexpected; leading cause of VI was uncorrected refractive error followed by cataract. I inferred from the paper that issues like spectacle correction were dealt with in an expedient manner, and referral for cataract surgery made. In all, an admirable reproducible study that should lead to better interventions and outcomes for patients.
well done.

Response: Thank you for the positive comments. We appreciate you taking time out to review our manuscript and provide comments to improve it further.